# Efficacy of a Task-Oriented Intervention for Children with a Dual Diagnosis of Specific Learning Disabilities and Developmental Coordination Disorder: A Pilot Study

**DOI:** 10.3390/children10030415

**Published:** 2023-02-21

**Authors:** Eugène A. A. Rameckers, Roche Crafford, Gillian Ferguson, Bouwien C. M. Smits Engelsman

**Affiliations:** 1Faculty of Rehabilitation Sciences, Hasselt University, B-3500 Hasselt, Belgium; 2Department of Rehabilitation Medicine, Research School CAPHRI, Maastricht University, 6200 MD Maastricht, The Netherlands; 3Adelante Rehabilitation Center, 6301 HA Valkenburg, The Netherlands; 4Department of Health & Rehabilitation Sciences, Faculty of Health Sciences, University of Cape Town, Cape Town 7925, South Africa; 5Faculty Health Sciences, Physical Activity, Sport and Recreation, North-West University, Potchefstroom 2520, South Africa

**Keywords:** DCD, co-morbidity, specific learning disability, task-oriented training, developmental coordination disorder, intervention, neuromotor task training

## Abstract

Background: Task-oriented approaches are recommended for children with developmental coordination disorder (DCD) to address deficits in motor performance and reduce activity limitations. Although this approach is used in several settings, the efficacy of these approaches in children with in dual-diagnosis of specific learning disabilities (SLD) and DCD is less widely known. This study aims to determine the effect of a group-based intervention based on neuromotor task training (NTT) principles on the motor performance of children aged 6–10 years with SLD/DCD. Methods: A pre-post-test controlled study design was conducted in children with a primary diagnosis of specific learning disabilities (SLD). DCD status was confirmed based on clinical assessment. Children scoring ≤16th percentile on the Motor Assessment Battery for Children 2nd Edition (MABC-2), who also presented with a functional motor problem, according to the MABC checklist were considered as having DCD. Children were allocated to the NTT intervention group based on teachers’ perceived notion of need and received two 45–60 min training sessions per week for nine weeks. Children allocated to the usual care (UC) group, received their planned occupational therapy and physical education. The MABC-2 was used to assess changes in motor performance. Outcome and Results: Our numbers confirm that it is crucial to identify the presence of motor coordination difficulties in children who have been diagnosed with SLD. A task-oriented training program based on NTT principles, presented in small groups, has a positive effect on the motor performance in learners with neurodevelopmental disorders and this effect was larger than in the usual care group. Conclusion and Implications: Although using a small group format in children with multiple neurodevelopmental disorders may be challenging for the therapists, it may be a way of delivering services to children in schools for special education. What this paper adds: Children with DCD plus LSD show improvement in their motor skills by performing group-based NTT in the school environment. Group-based NTT shows a significant improvement in the TSS score of the MABC-2 compared to usual care. Children with DCD plus SLD show equal effect sizes after NTT intervention as DCD without SLD.

## 1. Introduction

Research has shown that neurodevelopmental disorders, such as specific learning disorder (SLD), developmental coordination disorder (DCD), and attention deficit hyperactivity disorder (ADHD) co-occur frequently [1]. Specific learning disorder (SLD) is considered by the DSM-5 [2] and is a clinical neurodevelopmental disorder that hinders a child’s ability to learn or use specific academic skills. Signs of SLD may include difficulties with reading, writing, and arithmetic and are usually diagnosed once a child starts formal schooling [3]. Emerging evidence suggests that apart from poor academic functioning, many children with SLD also have motor problems [4,5]. Learning disorders affect 2–10 percent of the school-age population worldwide [1]. Importantly, among children who present with SLD, there is frequent co-occurrence of other neurodevelopmental disorders such as developmental coordination disorder (DCD) [6] and attention deficit hyperactivity disorder (ADHD) [1].

DCD is a primary motor coordination disorder in which the acquisition and execution of coordinated motor skills are substantially below expectation, given the child’s chronologic age and previous opportunities for skill learning and use [3]. Besides difficulty with participation in physical activities, poor playground interaction, and avoidance of physical activities [7,8,9], children with DCD experience difficulties with academic schoolwork. The nature and underlying aspects of these disorders are likely to be a multifaceted mix of motor and cognitive aspects, especially when DCD co-occurs with other neurodevelopmental disabilities such as SLD or ADHD.

The co-occurrence of DCD in learners with SLD can be linked to common areas of deficit leading to academic learning difficulties and also impairment in motor skill acquisition. Evidence shows a strong association between learning disorders, executive function problems, and motor development problems in children [10,11]. Neuroimaging techniques have shown that regions important to both motor and cognitive performance, such as the cerebellum, dorsolateral prefrontal cortex, and the connecting structures (including the basal ganglia) are co-activated during motor and cognitive tasks. There is also evidence to suggest that impairments in motor function co-occur with deficits in executive function [12,13,14]. Working memory specifically has been identified as an area of deficit in both learners with learning difficulties and those with motor problems [12,13]. According to another theoretical approach, co-occurrence would reflect partially common etiological bases, as proposed in the cerebellar hypothesis [15].

It has previously been suggested that at least 50 percent of children with learning difficulties have a co-morbid or co-occurring motor coordination disorder [1]. Although it is known that children with SLD present with motor problems, the severity and extent of motor problems and the impact of these motor problems on various areas, are less well understood [16]. Moreover, studies have shown that learners with motor coordination problems often experience reduced participation in social, physical, and leisure activities [17]. Participation is fundamentally important to children’s development as it enables them to develop the social and physical abilities required to thrive, as well as provide social-emotional well-being, a sense of meaning, and purpose in life [18]. These motor problems warrant as much attention as the learning difficulties, as the motor problems have a significant negative impact on activities of daily living such as self-care, play, leisure, schoolwork, and future vocational opportunities.

Learners with SLD and co-morbid DCD (SLD/DCD) are considered a unique group, thus interventions designed for children with motor coordination problems only (such as DCD), may not be applicable or lead to comparable effects. Learners with SLD present with poorer memory, difficulties with executive functioning, shorter attention span and decreased cognitive monitoring skills, which may impact the way they learn new motor skills [19]. It is therefore important to establish the efficacy of existing interventions in this special group. To our knowledge, no studies evaluated motor interventions in children with a dual diagnosis of DCD and LD. However, two studies investigated motor-based training in children with a single diagnosis of learning disorders. Emami Kashfi et al. investigated the effectiveness of a psychomotor intervention to improve motor skills in boys with LD (reading, writing, and/or mathematics) by focusing on laterality, balance, and coordination [20]. They compared the psychomotor training with supplemental regular educational services for their LD, but also with a group who received only regular educational services. Significant but similar improvements in the overall, fine- and gross motor skill performance (*p* < 0.05) were found for both groups receiving the psychomotor training with(out) additional regular educational services. The children that received only regular educational services did not improve on motor skill performance. Westendorp and coworkers reported that a motor learning program focusing on ball skills had a significantly larger improvement (*p* < 0.001) of object control performance in children with LD compared to a physical activity program comprising a regular physical education program targeting gymnastics, athletics, ball games, and other training.

In 2019, Blank et al. reports that intervention approaches from a task-oriented perspective yield strong effects in improving motor performance in DCD [21]. Task-oriented approaches tend to focus on motor performance, i.e., on learning particular motor skills, with attention to specific aspects of task performance that is causing the child difficulty. One example of a task-oriented approach is neuromotor task. Task-oriented approaches, based on NTT principles, whether delivered to individuals or in small group programs are both effective ways of teaching motor skills to children with DCD [22,23].

Although these interventions yield good results when used in children with DCD, it is not clear whether a task-oriented training intervention will influence motor performance in learners with SLD/DCD. Only one study so far examined the effect of NTT in children with learning disabilities, but that intervention was focused on handwriting skills only [24].

Studies with task-oriented interventions emphasize principles used to guide practice and the importance of providing feedback to enhance learning. For learners with SLD, giving clear instructions, feedback and constant motivation may be beneficial to support their learning. In NTT, the therapist guides the learner in the process of learning motor skills, while motivating and giving feedback to the learner. NTT provides a framework to help implement intervention tasks that are gradually made harder by increasing task demands (task loading) and promoting learning, within the constraints and limitations of the individual [21]. It has been shown that group-based NTT can be used effectively to treat children with DCD in areas of resource constraints, such as low-income schools [22]. It has also been found that group-based training produced similar gains in motor performance to individual-based training and group-based training may be the preferred treatment option due to the associated cost savings [22]. In their guideline, the European Academy for Childhood Disability (EACD) [21] suggests that task-oriented approaches are the recommended intervention strategies for learners with DCD and currently have the best cost-benefit. This study aims to contribute information on resource-efficient treatment methods (small group format) for overstrained educational systems struggling to provide the necessary care to children with developmental disabilities such as SLD and DCD. The main aim of this study is to determine the efficacy of an NTT-based task-oriented program on the motor performance of children aged 6–10 identified with SLD and DCD.

## 2. Methods

### 2.1. Research Setting

The research study was conducted in a public school for Learners with Special Education Needs (LSEN) located in Cape Town, South Africa. Learners at this school represent various socio-economic status (SES) groups (low, middle, and high). They are enrolled in the school if they have certain physical or learning barriers to learning in mainstream education. Learners are categorized according to their primary, followed by secondary diagnosis, e.g., learners with a primary diagnosis of SLD may also have ADHD as a secondary diagnosis. According to the school’s database, learners categorized according to primary diagnosis with SLD, currently make up 46 percent of the total school population. Learners with a primary diagnosis of ADHD make up 16 percent, learners with cerebral palsy 14 percent, learners with autistic spectrum disorder (ASD) 12 percent, learners with physical disabilities 8 percent, learners with a primary diagnosis of epilepsy 2 percent, learners who are primarily hearing-impaired 1 percent, learners with mild intellectual disability and learners with behavioral disorders as primary disability making up the remaining 1 percent of the total number. DCD is not officially recorded as a diagnosis in the school medical files.

The school offers a multidisciplinary team approach to support the holistic education of the children. The team includes speech therapists, occupational therapists, physiotherapists, and psychologists.

At this school, learners with SLD/ADHD, both with or without DCD, receive class-based occupational therapy (OT). OT focuses on fine motor, visual perceptual, cognitive skills, and processing deficits. Many of the learners with SLD/ADHD with motor problems are identified as requiring OT, based on an OT assessment. These learners may receive extra group therapy or are placed on the waiting list for individual OT.

Motor performance difficulties of children with DCD are often viewed as minor, thus not warranting intervention, compared with the needs of children with more severe impairments such as cerebral palsy. At the time of the study, learners with SLD /DCD at this school were not receiving any physiotherapy intervention thus making the need for such a program necessary.

### 2.2. Study Design and Sample Selection

A quasi-experimental design, with pre-and post-tests, was used for the intervention study, to explore the effects of this task-oriented NTT intervention in children with SLD and DCD. The study involved two groups of learners identified with SLD and DCD. A sample of convenience, consisting of children aged 6–10 attending this LSEN school was used to select children who met the inclusion criteria. All children aged 6–10 years, attending a special school in Cape Town, who presented with a primary diagnosis of SLD were recruited.

These learners were then assessed for DCD criteria, according to the DSM-5. The learners presented with motor skills below the expected norm for their age, scoring ≤16th percentile for their age on the MABC-2, indicating motor coordination problems.

MABC checklists were completed for each child by their class teachers to identify functional problems. According to the parent questionnaire, it was determined if the onset of symptoms was in the early developmental period. Children with intellectual disability (IQ below 70), severe visual impairment, cerebral palsy, muscular dystrophy, degenerative disorders, spina bifida, spinal cord injury (SCI), any syndromes, illness on the day of testing, acute fractures were not included in the study. Informed consent was obtained from parents and assent from the children to be assessed and participate in the study. See Figure 1.

No previous NTT studies aiming at the improvement of gross motor skills have been conducted among children with SLD/DCD. Therefore, the sample size calculations were based on a previous South African study in which the efficacy of NTT treatment determined by the change in MABC-2 scores was investigated [22]. The total sample size required in each arm of the intervention phase was calculated using the web-based statistical calculator. Accordingly, it was established that 36 participants were required to enter this two-treatment parallel-design study (18 participants per group) to detect a treatment change of two standard scores (SD = 2.5) on the MABC-2 at a 0.05 percent significance level with a 90 percent probability.

### 2.3. Outcome Measures

The M-ABC-2 is a standardized reliable, valid, and responsive instrument to measure children with motor impairments and assess the efficacy of treatment [25]. The specificity of the checklist is acceptable and it met the standards for validity and reliability [26]. The MABC-2 and Checklist are both standardized assessment tests designed to detect functional motor performance and motor skill impairment challenges in children. The MABC checklist investigates the effects of motor difficulties on activities of daily living, Standard Error of Measurement (SEM: 2 and 3 standard points), and Smallest Detectable Difference (four or more standard points) of MABC-2 total standard score.

Although there are no norms for MABC-2 in South Africa, it has been used in South African studies with a focus on motor performance and developmental coordination disorder using the Dutch norms.

### 2.4. Procedure

Ethical approval was obtained by the Human Research Ethics Committee (HREC) of the University of Cape Town (UCT HREC Ref 426/2016). The principal and the Western Cape Education Department granted permission to conduct research at the school.

Consent forms were distributed to all learners who met the age bracket of the inclusion criteria (N = 85). Fifty-eight parents agreed to participate in the study and each child provided consent. The remaining 27 parents either did not return consent forms (n = 26) or indicated that they were not interested in participating in the study (n = 1). Learners were assessed by an independent researcher to determine the scores on the MABC-2. Functional motor problems were identified at baseline by the teacher and the parent questionnaire. Thirty-six learners (N = 36) were identified as meeting the criteria for DCD and were enrolled in the study (see Figure 1). Information on the diagnosis of the included children is summarized in Table 1.

To be included in the intervention, learners had to score ≤ 16th percentile (total standard scoreTSS) on the MABC-2 and present with a functional motor problem as determined by the MABC Checklist (teacher) and questionnaire (parent), therefore, have a concurrent diagnosis of DCD as per the DSM-5 criteria. The learners were then divided equally into an intervention (n = 18) or control (n = 18) group. Allocation to the intervention or control group was performed by the teachers. Teachers allocated learners to the intervention group based on their perceived impression of who “required therapy the most” based on observed functional and motor coordination problems. Teachers allocated learners to groups based on a learner’s need for therapeutic intervention in the area of motor difficulties. The remaining learners were then allocated to the control group. Thus, all learners in the intervention and control group presented with motor coordination problems according to the MABC-2 test scores, but the learners whom teachers felt presented with functional motor coordination difficulties affecting classroom functioning were allocated to the intervention group. The control group received the usual care.

### 2.5. Intervention

The task-oriented intervention was developed based on the literature and in consultation with experts in the field of NTT (BSE and ER), who were not involved in delivering the intervention or pre and post-tests. The therapist responsible for implementing the intervention received training before starting the study and was blinded to the pre-intervention scores of learners.

The principles of NTT are based on task analysis, thus breaking a task down into parts. This is the basis on which skills are taught in the task-oriented approach and enables the focus to fall on the main problem areas in the task. Task analysis, being a key principle of NTT, incorporates planning, execution, and evaluation to be able to adapt the task to make it achievable for the child and therefore facilitate learning. Furthermore, skills are learned progressively through task loading, changing spatial and temporal constraints of the task, and by combining tasks, dependent on the learning stage a child has reached for a specific skill [27]. The program content was developed based on the learners’ areas of difficulty.

Sessions consisted of various activities set up as stations. Activities included components of soccer, netball, and basketball, variations of tagging games, skipping with a rope, and other popular games organized as workstations. Children participated under the therapist’s guidance, who manipulated aspects of the environment and tasks as needed. One therapist (trained physiotherapist) ran each intervention group along with a trained assistant (not a therapist) to assist in each group. The groups had a ratio of 2:1 (children to adult supervisor).

The intervention took place over nine weeks, with two sessions per week each lasting between 45 and 60 min. Each group consisted of four children. For more details on the content of the intervention protocol, see Appendix A (Table A1).

Usual care (UC) included basic physical education presented by class teachers as well as occupational therapy (OT) either in the class group, small group, or individual format.

The UC group received regular physical education one time a week, including gymnastic exercises and sports. OT one time a week consisted of training in fine motor activities and sensorimotor therapy.

The intervention group also continued with physical education as well as any OT classroom activities, as no learner was excluded from other activities based on their participation in the study.

Pre- and post-testing (MABC-2) was performed by research assistants who were trained and experienced with administering the MABC-2 and independent of the study. These research assistants were blinded to participants’ allocation to groups.

### 2.6. Data Analysis

The Shapiro–Wilk test and Levene’s Test for Equality of Variances for normality were used to determine whether assumptions were met for parametric analysis. Comparisons between the age in the two intervention groups were assessed using independent *t*-tests and the Chi-square test was used to compare the frequency of boys and girls.

To test the effects of the intervention, the General Linear Model was used for MABC-2 measures. Time of assessment (i.e., pre- and post-intervention) was used as the within-subjects factor and group (NTT/Usual care) as the between-subjects factor.

Given the small group size (n = 18), Wilcoxon paired sample *t*-test was used as a post hoc test for pre-post-comparison within NTT and the Usual care group.

Effect sizes (d) were calculated to determine the practical significance of these differences; d-values greater than 0.5 were taken to indicate a moderate effect and values greater than 0.8 were taken to indicate a large practical significance [28]. Statistical significance was noted if *p* < 0.05. All statistical analyses were performed using the Statistical Package for the Social Sciences (SPSS Inc., Armonk, NY, USA version 26).

## 3. Results

The intervention group consisted of 18 children (14 boys and four girls), with a mean age of 9.14 years (SD = 1.25) and the usual care consisted of 18 children (13 boys and five girls), with a mean age of 9.51 years (SD = 1.12). About 18 children were also diagnosed with ADHD (see Table 1). No significant difference was found between the groups in terms of mean age at pre-test (t = 0.91, df = 34, *p* = 0.37) or gender distribution (Chi^2^ = 0.148, df = 1, *p* = 0.7) (Table 1).

Mean attendance for the 18 participants in the intervention group was 15 sessions out of a maximum of 18 sessions (SD = ±1.28). There were no deviations from the protocol or adverse reactions during and after the intervention program.

The analysis of the overall motor performance changes (Table 2), as reflected by the mean TSS of the MABC-2 for the total group, revealed a significant difference in motor performance (main effect time *p* = 0.003). The MABC-2 (n = 36) pre-test percentile scores ranged between the 0.1st percentile and the 16th percentile, and the post-test scores ranged from the 0.1st percentile to the 37th percentile.

The NTT group improved more than the UC group on the Total MABC-2 score, which was confirmed by a significant interaction (*Time* × *group p* = 0.041).

The analysis of balance scores also indicated a significant difference over time (*p =* 0.04), but the interaction was not significant (time × group *p* = 0.27).

The result of Wilcoxon paired sample *t*-tests indicated the mean TSS of the NTT group improved significantly over the intervention period (*p =* 0.002) yielding a medium effect size (d = 0.66). The balance component score in the NTT group also showed a significant improvement (mean difference *p =* 0.02), though yielding a rather small effect size (d = 0.19). The control group did not show any significant changes over the intervention period while receiving usual care (Table 3).

At an individual level, 50% of the children (9/18) in the NTT group improved more than the SEM while this was 18% (5/18) for the usual care group (See Table 4). Of the nine children who improved after NTT training four had a dual diagnosis (SLD/DCD) and five had multiple diagnoses.

## 4. Discussion

The main aim of this study was to determine the efficacy of an NTT-based task-oriented program on the motor performance of children aged 6–10 identified with co-occurring developmental disorders (SLD/DC). Results show that task-oriented intervention can improve motor performance in children with SLD/DCD.

All the children in this study went through an extensive procedure for the application to special needs schools and have confirmed learning disabilities, which makes it a special group not studied so far. A defining feature of many neurodevelopmental disorders is their frequent association with motor coordination problems. This study showed that only 9 out of the initially eligible 49 children with SLD presented scores in the normal range of the MABC-2 and 36 children presented with motor problems (73%). This is significantly higher than the percentage suggested in the literature. In the group of learners included in the study (n = 36), 92% percent scored below the fifth percentile on the MABC-2 pre-test. Furthermore, of these children 53% scored at or below the 0.5th percentile on the pre-test, indicating severe motor coordination problems [26]. These numbers confirm that it is crucial to identify the presence of motor coordination difficulties in children who have been diagnosed with SLD. Moreover, 16 learners in our study had only the dual diagnosis (SLD/DCD) while the 20 other learners had an additional diagnosis, mainly ADHD (50%). The comorbidity of attention deficit hyperactivity disorder (ADHD) in children with SLD varies from about 10% to as high as 60% depending on the sample, so our numbers are in concordance with the literature [29]. Children with ADHD are reported to meet the criteria for developmental coordination disorder (DCD) in approximately 50% [30], which was recently confirmed in a study by Farran and colleagues who reported a motor deficit was observed in 47% of their ADHD sample [29]. As reported in many studies and as shown in our study, a single disorder seems to be the exception while having different diagnoses or co-occurrent disorders or overlap between the disorders is the most prevalent situation in children with neurodevelopment disorders [31].

### 4.1. Effect of the Intervention

Task-oriented intervention based on NTT principles has previously been shown to have positive effects on populations of children with DCD [21]. NTT focuses directly on teaching the skills that a child needs to master to perform functional activities and to transfer acquired skills to daily life performance. This is the first study investigating the intervention in a special population of learners with SLD/DCD. We hypothesized that the NTT-based program would have a positive effect on motor performance in learners with SLD/DCD because it has shown positive effects in a similar population group without comorbidity. However, knowing the constraints in learning, executive function, sustained attention, visuospatial working memory, and impulse control in the current group and the severity of the motor coordination problems, we did not expect similar improvements as described in children with DCD without SLD.

The study found a medium to small effect of NTT in the SLD/DCD group. The NTT group showed a significant improvement in overall motor performance (TSS) and balance scores after the nine-week intervention. Although the group size was small, having multiple diagnoses (mainly SLD/DCD/ADHD) compared to SLD/DCD did not seem to negatively impact the results. The improvement may be attributed to the principles underpinning NTT-based interventions. This includes guided discovery to facilitate implicit learning of task components with positive feedback to support learning, as well as focusing on planning, execution, and evaluation to be able to adapt the task to make it achievable for the child and therefore facilitate learning. NTT also has broad advantages in the SLD co-morbid population as these learners often present with a low attention span, difficulty in learning new concepts, and difficulty following instructions. The NTT approach acknowledges that learning and skill acquisition are the strongest when the learner understands the meaning of the exercise and finds the task to be useful or relevant to his or her life, and is thus valid in the child’s environment with the support of parents and teachers [27]. In this study, the exercises chosen were useful, relevant, and valid. Learners practice activities that are representative of daily life, on the playground, or at home, and which they wanted to improve. This approach enables the child to interact with the environment resulting in acquiring new or improved motor function. For example, learners practiced components of soccer, which is a popular game often played at school and in communities. In the NTT-based approach, learners participate in activities that are familiar tasks but adapted to their motor skill level by the therapist. Task-oriented approaches such as NTT are regarded as active approaches to motor learning with low cognitive demand, therefore making it very suitable for this population. According to teaching principles, low cognitive demand tasks involve stating facts, following known procedures, and solving routine problems. It seemed that the approach was feasible to use in children with comorbidities.

Although a significant improvement in overall motor performance was observed for learners who received NTT intervention, the effect size was medium to small. This could be attributed to the fact that learners with SLD have general difficulties in areas of learning. According to the DSM-5, SLD is considered to be a type of neurodevelopmental disorder that hinders the learner’s ability to learn or use specific academic skills (e.g., reading, writing, or mathematics), while DCD is defined as a failure to have acquired age-appropriate motor actions despite adequate opportunity and practice. Thus, the limited ability to learn and automatize may be a common factor and may also have affected the rate of motor skill acquisition in the current study [32]. Considering that children with SLD are significantly impaired in a large range of processes known to be dependent on the cerebellum, such as executive functioning, memory, learning, attention, visuospatial regulation, language, and motor skills. It has been suggested that cerebellar dysfunction could constitute a common causal factor in comorbid SLD and DCD [33]. Importantly, the cerebellum has also been mentioned to be implicated in ADHD, both structurally [33] and functionally [34].

Another factor explaining the medium to small effect size can be the duration of the training period (9 weeks), related to the automation of the trained skills. It is expected that a longer duration will increase the possibility for a higher level of automation, however, will decrease the practicability and feasibility of the training at a school. The comprehensiveness of the NTT treatment can be an important reason explaining the presented effect. This is in line with the evidence on dose-intensive studies, as reported in the field of children with Cerebral Palsy [35] and children with motor deficits [36], showing that increased dosage of intervention is showing higher effects compared to low dosage intervention.

Although the NTT group showed significantly more improvement than the usual care group, still only half the children improved more than the SEM of the total MABC-2 score. It is important to find ways to help the non-responding half of the group. Would extending the training time be enough? Or should the frequency of training be increased within the weeks? Or should the instruction and loading of the tasks be even more simplified and adapted for this not (yet) responding group?

### 4.2. Usual Care

The UC group did not show any significant change in any area (TSS, Balance, A&C, MD) over the nine weeks (Table 3). These results may indicate the need for further research to determine whether UC in this population has any effect on motor performance. It is worthwhile for future studies to examine, which children do improve and which do not because 5 of the 18 children did show improvement above the SEM.

One of the weaknesses of this study is the chosen design, not being an RCT. However, to explore the first effects of the NTT in this new group of children, a quasi-experimental design, with pre-and post-tests in the real school environment of the children is the first step to an RCT design. The results are strong enough to explore the possibility to perform an RCT design in the future.

Another weakness is that we did not measure the specific executive functions of the children participating in this study. This needs to be added in future research to have a specific profile of the children regarding their executive function ability.

In summary, the NTT intervention resulted in statistically significant and clinically important improvements in motor performance (TSS and Balance) from the pre- to post-intervention test. Response to intervention was 50%. This finding concurs with findings of reviews of interventions for learners with DCD where large effect sizes for task-oriented approaches were reported [20,22]. In contrast, the usual care group showed no statistically significant change.

### 4.3. Recommendations for Future Research

There are several weaknesses in the methodology employed in our study.

The effect size of the NTT program was found to be smaller in this study than in other studies investigating task-oriented approaches. This could be because of the complexity of the neurodevelopmental disorders in this study. The optimal frequency of training for coordination purposes in learners with SLD/DCD has not been established. The frequency of the intervention was based on previous studies using task-oriented approaches in learners with DCD [20,22]. Further research is needed to determine if learners with co-morbid DCD would benefit from higher frequency intervention. In this study, the intervention was presented in small groups. It has been found that group settings offered more opportunities for social interaction, motivated children to compete with each other, and contributed to a stronger sense of ability as a result of successful performance in front of other learners in the group [27]. However, guiding a small group of children with diverse limitations through an intervention is challenging for supervisors. An extra person (for instance a teaching assistant) next to the therapist is needed to get the training well organized. Moreover, time management in a school context and collaboration with the teachers are prerequisites to making the program feasible. Further research is needed to determine if learners with co-morbid DCD will show greater improvement when treated in pairs or individually. The effects of small group training on social skills and behavior still need to be studied.

In this study, the effect of an NTT program was investigated in an SLD/DCD population, making this study different from previous studies investigating NTT in learners presenting with DCD. Further research is needed to investigate the effect of a task-oriented approach, like NTT, on all domains of the ICF, including participation, and body structure and function, the effect of the improvement in physical activity in the home and school environment in terms of behavioral profiles as well as the quality of life, therefore looking at the personal and environmental domains of the ICF. This study also confirms the crucial need to assess motor skills among children with SLD and develop ways to incorporate motor skill training into their treatment plans.

## 5. Conclusions

The literature on children with neurodevelopmental disorders acknowledges that they are a heterogeneous group concerning the profile of perceptual–motor problems. Their clinical picture is complicated because of the frequent co-morbidity of developmental disorders. Whether children with SLD, DCD, or ADHD and children with a combination of any of these developmental disorders differ from each other in terms of the response to intervention has received little attention. Yet, it is clear that the outcome of such studies is of relevance in establishing the validity of separate treatment approaches for these combinations of common developmental disorders. The results of this study showed that a nine-week task-oriented program had a positive effect on motor performance in learners with neurodevelopmental disorders. This study provides evidence for effective interventions that can be implemented in schools, specifically LSEN schools, to improve motor performance in learners with SLD and co-occurring DCD. The study also demonstrated that NTT can be presented in a small-group format. This makes this approach cost-and-time-effective in an under-resourced setting where there is a serious need for intervention in children with motor problems.

One of the most marked limitations of the study was that conducting a randomized controlled trial was not possible. Due to the teachers allocating learners to groups based on perceived motor performance needs, selection bias could have occurred. Randomization of learners would have led to better quality research. Learners in both the NTT and usual care group continued with their regular OT and physical education with their class and therefore the effect of OT and physical education could not be excluded in this study but was used as the control intervention. Only immediate post-intervention effects were investigated in this study; therefore, retention of effect was not investigated. Further research is needed to determine the retention of skills in this special population.

## Figures and Tables

**Figure 1 children-10-00415-f001:**
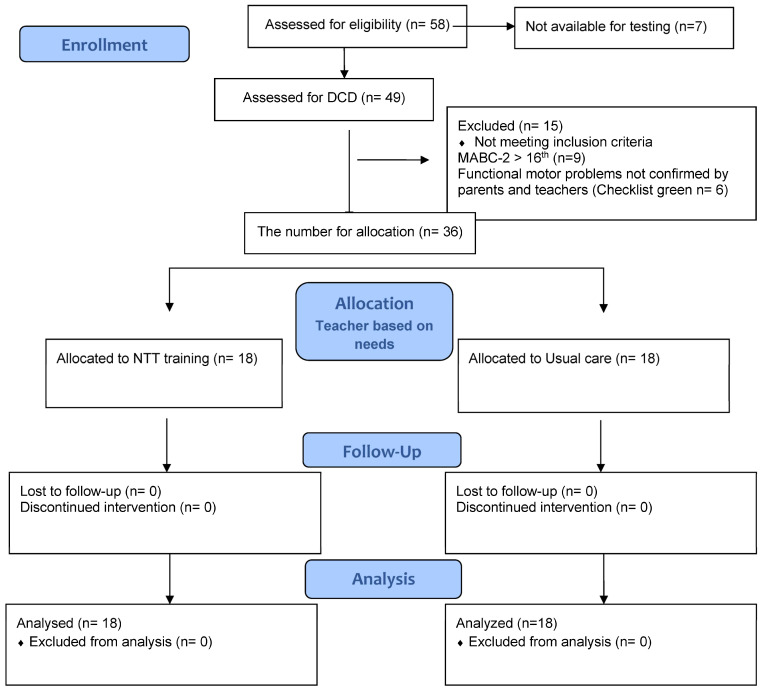
Flowchart of children’s recruitment process of the study.

**Table 1 children-10-00415-t001:** All children included in the study had multiple diagnoses. SLD: Specific learning disorder, DCD developmental coordination disorder, ADHD attention deficit hyperactivity disorder, PDD pervasive developmental disorder, NTT Neuromotor Task Training.

Diagnosis	A Usual Care Group (N = 18)	NTT Group (N = 18)
SLD/DCD	N = 7	N = 9
SLD/DCD/ADHD	N = 10 (all on medication)	N = 8 (6 on medication)
SLD/DCD and additional problems	1 PDD	1 Orthopedic chest problem

**Table 2 children-10-00415-t002:** Results of the repeated measures, the main effect of time (pre and post), and interaction effect of time by group (NTT and usual care) with F value, *p*-value, and effect size on MABC-2 standard scores (TSS = total standard score; MD = manual dexterity; A&C aiming and catching; and balance).

	Time [1, 34]			Time × Group [1, 34]		
MABC-2	F-Value	*p*-Value	Eta Squared	F-Value	*p*-Value	Eta Squared
TSS	10.13	0.003	0.23	4.50	0.041	0.12
MD	1.08	0.31	0.03	2.67	0.11	0.07
A&C	0.02	0.88	0.001	0.59	0.48	0.02
Balance	4.51	0.041	0.12	1.24	0.27	0.035

**Table 3 children-10-00415-t003:** Mean (SD) pre and post-intervention standard scores on MABC-2 for UC (n = 18) and NTT group (n = 18). (t-values, *p*-values, and Cohen D effect size). TSS: Total standard score; MD: manual dexterity; A&C: aiming and catching. Z: z-value Wilcoxon; *p*: *p*-value; d: Cohen d; UC; usual care; NTT: neuromotor task training.

	UC				NTT					
	Pre	Post	Z	*p*	D	Pre	Post	Z	*p*	D
TSS	2.83 (2.07)	3.11 (2.08)	−0.781	0.44	0.13	3.22 (2.05)	4.61 (2.15)	−3.03	0.002	0.66
MD	3.56 (2.12)	3.39 (2.38)	0.443	0.66	−0.07	3.78 (2.41)	4.78 (2.86)	−1.68	0.09	0.38
A&C	4.39 (2.52)	3.89 (3.18)	0.599	0.55	−0.17	5.39 (2.68)	5.72 (2.67)	−0.486	0.63	0.12
Balance	4.89 (2.89)	5.44 (2.87)	−0.719	0.47	0.19	5.89 (2.65)	6.67 (5.17)	−2.27	0.02	0.19

**Table 4 children-10-00415-t004:** Number of children that changed more than the Standard Error of Measurement (SEM: 2 and 3 standard points) and Smallest Detectable Difference (4 or more standard points) of MABC-2 total standard score per group, UC = usual care, NTT = neuromotor task training.

Group			Individual Change	
		Less than SEM	Above SEM	Above SDD
UC	Count Number (%)	13 (72.2)	5 (17.8)	0 (0)
NTT	Count Number (%)	9 (50)	8 (44.4)	1 (5.6)

## Data Availability

Research data can be requested by sending e-mail to UCT G. Fersuson. gillian.ferguson@uct.ac.za.

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
