# Peer review of "Efficacy of a Task-Oriented Intervention for Children with a Dual Diagnosis of Specific Learning Disabilities and Developmental Coordination Disorder: A Pilot Study"

_children, 2023, doi:10.3390/children10030415_

Round 1

Reviewer 1 Report

The present article discusses approaches to individuals with developmental disabilities and coordination disorder. I appreciate the focus of the article, which reflects the rather topical issue of these diagnoses. The age range of the probands is appropriately chosen, which also corresponds to the assessment domain of the applied MABC-2 test. In this case, the authors follow a very logical and systematic approach. The introduction and methods are clearly described and target the chosen topic. I do not find too much general information here, which is more burdensome than providing specific information. I must say that this is a fairly well done article. The problem I have is primarily with the results, I don't know if this is a bug within the uploading of the paper, but line 318 mentions some sort of error. I would have presented the results better in the form of clear tables followed by descriptions. the authors give the p level values, which is fine. But the interpretation of the data is very vague. I also miss the rationale why the T-test was chosen for 18 probands? What are the research objectives? The research problem and possibly the hypothesis? The results are meant to test a hypothesis that is not stated here. The null and alternative hypotheses are also missing. I ask the authors to clearly add statistical procedures. In the case of conclusions, I would suggest looking for the addition of a suitable source that points to, for example, comprehensive rehabilitation approaches. This will help the authors to generalise the results and highlight the different approaches within the therapy of these individuals. It may be appropriate to reflect special education or occupational therapy approaches. I have searched by keywords for some articles to illustrate, please reflect on them in the conclusion. This will conclude an otherwise quite well developed article.

1. https://www.webofscience.com/wos/woscc/full-record/WOS:000794373400001

2. https://www.webofscience.com/wos/woscc/full-record/WOS:000839346100001

From the given articles, it is possible to inform the reader about the possibilities of comprehensive approaches within the framework of cooperation between the different disciplines of the so-called helping professions. I also recommend focusing on the use of more up-to-date resources. In the reference list you have sources from 1999,1996,2001 etc. It doesn't seem to me that the articles in question are anywhere more up to date and since your article is not built directly on a comparison of historical and current trends, try to recreate the oldest ones. If reflecting the changes mentioned, I would be inclined to publish the article in question.

Author Response

We thank the reviewers for their constructive feedback to improve the paper.

Both reviewers seemed not to have access to the added tables and figures, explaining many questions addressed by the reviewers. I added all tables and figures in the text to illustrate the results completely and I hope they will clarify the results, presented in the text.

Reviewer 1

The present article discusses approaches for individuals with developmental disabilities and coordination disorders. I appreciate the focus of the article, which reflects the rather topical issue of these diagnoses. The age range of the probands is appropriately chosen, which also corresponds to the assessment domain of the applied MABC-2 test. In this case, the authors follow a very logical and systematic approach. The introduction and methods are clearly described and target the chosen topic. I do not find too much general information here, which is more burdensome than providing specific information. I must say that this is a fairly well done article.

Answer:

We thank the reviewer for the compliments

The problem I have is primarily with the results, I don't know if this is a bug within the uploading of the paper, but line 318 mentions some sort of error.

Answer:

This seems indeed an uploading error, the location of a table was described on this line, but not presented.. In the revision, this have been checked in detail.

I would have presented the results better in the form of clear tables followed by descriptions. The authors give the p-level values, which is fine. But the interpretation of the data is very vague.

Answer:

In the revised version all tables and figures are embedded in the results section and the details as expected will be presented in the tables.

I also miss the rationale why the T-test was chosen for 18 probands.

Answer:

We expect the reviewer is referring to this sentence” Given the small group size (n=18), Wilcoxon paired sample t-tests were used for pre-post-comparison within the groups”. After performing the General Linear Model interactions between time and group became evident. Therefore post hoc tests were performed within the subgroups (NTT/Usual care). It is standard to use nonparametric tests when assumptions of parametric tests cannot be achieved, or the sample size is small. Thus, we used Wilcoxon signed-ranks test the non-parametric equivalent of the paired t-test. Thank you for pointing out that we didn’t mention this test was done for post hoc purposes. We have added this to the text.

What are the research objectives? The research problem and possibly the hypothesis? The results are meant to test a hypothesis that is not stated here. The null and alternative hypotheses are also missing.

Answer

We reported in the introduction the aim of the study and we added the hypothesis we tested.

We changed to text in the introduction:

The main aim of this study was to determine the efficacy of an NTT-based task-oriented program on the motor performance of children aged 6–10 identified with SLD and DCD. It was hypothesized that the changes over time would be larger in the NTT group than in the usual care group. Effect sizes were calculated to determine the practical significance of the change.

In the discussion we report the hypothesis again when discussing the effects of intervention:

We hypothesized that the NTT-based program would have a positive effect on motor performance in learners with SLD/DCD because it has shown positive effects in a similar population group without comorbidity. However knowing the constraints in learning, executive function, sustained attention, visuospatial working memory, and impulse control in the current group and the severity of the motor coordination problems, we did not expect similar improvements as described in children with DCD without SLD.

I ask the authors to clearly add statistical procedures.

Answer:

In the data analysis section, we added information (highlight) about the choice of the tests

The Shapiro-Wilk test and Levene's Test for Equality of Variances for normality were used to determine whether assumptions were met for parametric analysis. Comparisons between the age in the two intervention groups were assessed using independent t-tests and the Chi-square test was used to compare the frequency of boys and girls.

To test the effects of the intervention, the General Linear Model was used for MABC-2 measures. Time of assessment (i.e. pre- and post-intervention) was used as the within-subjects factor and group (NTT/Usual care) as the between-subjects factor.

Given the small group size (n=18), Wilcoxon paired sample t-test was used as a post hoc test for pre-post-comparison within NTT and the Usual care group.

Effect sizes (d) were calculated to determine the practical significance of these differences; d-values greater than 0.5 were taken to indicate a moderate effect and values greater than 0.8 were taken to indicate a large practical significance 38. Statistical significance was noted if p < 0.05. All statistical analyses were performed using the Statistical Package for the Social Sciences (SPSS Inc., version 26).

In the case of conclusions, I would suggest looking for the addition of a suitable source that points to, for example, comprehensive rehabilitation approaches. This will help the authors to generalize the results and highlight the different approaches within the therapy of these individuals. It may be appropriate to reflect special education or occupational therapy approaches. I have searched by keywords for some articles to illustrate, please reflect on them in the conclusion. This will conclude an otherwise quite well-developed article.

  1. https://www.webofscience.com/wos/woscc/full-record/WOS:000794373400001
  2. https://www.webofscience.com/wos/woscc/full-record/WOS:000839346100001

From the given articles, it is possible to inform the reader about the possibilities of comprehensive approaches within the framework of cooperation between the different disciplines of the so-called helping professions.

Answer:

I thank the reviewer for the references and the need to elaborate in the discussion on the comprehensive treatment.

We added the paragraph:

The comprehensiveness of the NTT treatment can be an important reason explaining the presented effect. This is in line with the evidence on dose-intensive studies, as reported in the field of children with Cerebral Palsy39 and children with motor deficits40, showing that increased dosage of intervention is showing higher effects compared to low dosage intervention..

I also recommend focusing on the use of more up-to-date resources. In the reference list you have sources from 1999,1996,2001 etc. It doesn't seem to me that the articles in question are anywhere more up to date and since your article is not built directly on a comparison of historical and current trends, try to recreate the oldest ones. If reflecting the changes mentioned, I would be inclined to publish the article in question.

Answer:

Thank you for the recommendation. We updated the oldest papers, where needed.

We replaced the studies:

(1) Geuze, R. H.; Jongmans, M.; Schoemaker, M.; Smits-Engelsman, B. Developmental coordination disorder. Hum Mov Sci 2001, 20 (1-2), 1-5. DOI: 10.1016/s0167-9457(01)00026-4.

(4) Margari, L.; Buttiglione, M.; Craig, F.; Cristella, A.; de Giambattista, C.; Matera, E.; Operto, F.; Simone, M. Neuropsychopathological comorbidities in learning disorders. BMC Neurology 2013, 13 (1), 198, journal article. DOI: 10.1186/1471-2377-13-198.

(6) Al-Mamari, W. S.; Emam, M. M.; Al-Futaisi, A. M.; Kazem, A. M. Comorbidity of Learning Disorders and Attention Deficit Hyperactivity Disorder in a Sample of Omani Schoolchildren. Sultan Qaboos University Medical Journal 2015, 15 (4), e528-e533. DOI: 10.18295/squmj.2015.15.04.015 PMC.

(7) Lingam, R.; Golding, J.; Jongmans, M. J.; Hunt, L. P.; Ellis, M.; Emond, A. The association between developmental coordination disorder and other developmental traits. Pediatrics 2010, 126 (5), e1109-1118. DOI: 10.1542/peds.2009-2789

(10) Kaplan, B. J.; Dewey, D. M.; Crawford, S. G.; Wilson, B. N. The term comorbidity is of questionable value in reference to developmental disorders: data and theory. Journal of learning disabilities 2001, 34 (6), 555-565. 

(27) Niemeijer, A. S.; Smits-Engelsman, B. C. M.; Schoemaker, M. M. Neuromotor task training for children with developmental coordination disorder: A controlled trial. Developmental Medicine and Child Neurology 2007, 49, 406–411.

(29) Niemeijer, A. S.; Smits-Engelsman, B. C.; Schoemaker, M. M. Neuromotor task training for children with developmental coordination disorder: a controlled trial. Dev Med Child Neurol 2007, 49 (6), 406-411. DOI: 10.1111/j.1469-8749.2007.00406

Reviewer 2 Report

I carefully reviewed the manuscript which verifies the effect of Neuromotor Task Training (NTT) in a group of children with a dual-diagnosis of Specific Learning Disabilities (SLD) and Developmental Coordination Disorder (DCD). The sample was divided into two groups: one received the NTT intervention while the other had the Usual Care (UC). The results show that the NTT had a positive effect on motor learners with SLD and DCD.

The study deserves attention mainly for two reasons: 1. It underlines the opportunity to consider motor problems within the SLD, in line with the comorbidity among neurodevelopmental disorders; 2. It offers findings to better think about variables that must be considered within the intervention plans dedicated to children with a complicated clinical profile.

This study is worthily of publication, however, at the present form, the manuscript needs some improvements. 

I point out some suggestions below here:

Abstract

It is fine.

What this paper adds

Line 43, the acronym “TSS” is not explained.

1.     Introduction

Line 54, after “the DSM-5” substitute “or” with “to”.

Line 7, “the co-occurrence of DCD in learners with SLD can possibly be linked to common areas of deficit leading to academic learning difficulties and also impairment in motor skill acquisition…” I would suggest to say something like “Beyond each specific neurocognitive mechanism which regulates academic and motor learning (reading, spelling and motor skills), common area of deficit also exists when DCD occurs in learners with SLD”.

Line 83, the sentence states that “at least 50 per cent of children with learning difficulties have co-morbid or co-occuring motor coordination disorder” by referring the study of Fortes et al. (2016) who presented findings from a Brazilian sample. I would suggest not generalized to the worldwide population and, maybe, it would be useful to refer to Cleaton M. A. M., Kirby A. (2018) Why Do We Find it so Hard to Calculate the Burden of Neurodevelopmental Disorders? Journal of Childhood & Developmental Disorders. Vol. 4, 1-20. DOI: 10.4172/2472-1786.100073; http://childhood-developmental-disorders.imedpub.com/archive.php

2.     Method

Unlikely, I do not have access to figure and tables so I did not control figure and tables.

Line 196, Reference 33 is already cited (number 26).

Line 198, reference is not correct.

Line 210, something is missing in the sentence “Standard Error of Measurement …)

In session 2.5 Intervention I would suggest to clarify better what saying at line 263, if I understood well the 18 children, who did NNT intervention, were trained in couple or maximum in four children. It is not crystallin clear to me.

From line 268 to 272 a meagre description is provided for children who did Usual Care. Could the authors provide more details related to the kind of proposed activities in UC?

3.     Results

I restate that I do not have the possibility to see tables, however, text at line 305-308 has something missing. At line 308, the acronym “TSS” is not explained.

Probably, the sentence from line 318 to 320 should be insert after line 313.

4.     Discussion

In general discussion presents deep thoughts in relationship to the findings however, references, used to support the reasonings, need to be checked, for example, line 363 reference 44 is already reported as number 19.

From line 405 to 407, the authors speculate that the reason for the effect size medium to small after the NTT intervention could be due to the general difficulties in area of learning and its automation. However, another possibility could be the necessity of longer period of training to stabilize motor acquisitions.

From line 427, weaknesses of this study are illustrated. Since several times executive function problems are held to be cause of the complicated clinical profile of children with SLD/DCD, I would suggest to mention that no measurements related to executive function were reported in this study.

At line 437, reference numbered 44 is reported before with number 19.

At line 462, reference of ICF is not indicated.

Reference needs to be checked from number 33 there are some mistakes:

-       reference 33 is already cited as 26;

-       reference 34 is not completed;

-       after reference 36, the paper of Smits-Engelman et al. (2013) is already cited as 23 so it needs to be cancelled;

-       reference 44 is cited as 19;

-       reference 48 and 50 need to start a new line

Author Response

We thank the reviewers for their constructive feedback to improve the paper.

Both reviewers seemed not to have access to the added tables and figures, explaining many questions addressed by the reviewers. I added all tables and figures in the text to illustrate the results completely and I hope they will clarify the results, presented in the text.

Reviewer 2

I carefully reviewed the manuscript which verifies the effect of Neuromotor Task Training (NTT) in a group of children with a dual-diagnosis of Specific Learning Disabilities (SLD) and Developmental Coordination Disorder (DCD). The sample was divided into two groups: one received the NTT intervention while the other had the Usual Care (UC). The results show that the NTT had a positive effect on motor learners with SLD and DCD.

The study deserves attention mainly for two reasons: 1. It underlines the opportunity to consider motor problems within the SLD, in line with the comorbidity among neurodevelopmental disorders; 2. It offers findings to better think about variables that must be considered within the intervention plans dedicated to children with a complicated clinical profile.

This study is worthily of publication, however, at the present form, the manuscript needs some improvements. 

I point out some suggestions below here:

Abstract

It is fine.

Answer:

Thank you

What this paper adds

Line 43, the acronym “TSS” is not explained.

Answer:

We explained the Total Standard Score  (TSS)

  1. Introduction

Line 54, after “the DSM-5” substitute “or” with “to”.

Answer:

We changed this into “to”

Line 7, “the co-occurrence of DCD in learners with SLD can possibly be linked to common areas of deficit leading to academic learning difficulties and also impairment in motor skill acquisition…” I would suggest to say something like “Beyond each specific neurocognitive mechanism which regulates academic and motor learning (reading, spelling and motor skills), common area of deficit also exists when DCD occurs in learners with SLD”.

Answer:

Thank you. We accepted the changes and added the proposed text.

Furthermore we added extra text in the introduction to enlighten  the complex motor problems in children with learning disabilities

“” However, two studies investigated motor-based training in children with a single diagnosis of learning disorders. Emami Kashfi et al. investigated  the effectiveness of a psychomotor intervention to improve motor skills in boys with LD (reading, writing, and/or mathematics) by focusing on laterality, balance, and coordination22. They compared the psychomotor training with supplemental regular educational services for their LD, but al-so with a group who received only regular educational services. Significant but similar improvements in the overall, fine- and gross motor skill performance (p<0.05) were found for both groups receiving the psychomotor training with(out) additional regular educational services. The children that received only regular educational services did not im-prove on motor skill performance. Westendorp and coworkers reported that a motor learning program focusing on ball skills had a significantly larger improvement (p<0.001) of object control performance in children with LD compared to a physical activity program comprising a regular physical education program targeting gymnastics, athletics, ball games, and other training5.

Line 83, the sentence states that “at least 50 per cent of children with learning difficulties have co-morbid or co-occuring motor coordination disorder” by referring the study of Fortes et al. (2016) who presented findings from a Brazilian sample. I would suggest not generalized to the worldwide population and, maybe, it would be useful to refer to Cleaton M. A. M., Kirby A. (2018) Why Do We Find it so Hard to Calculate the Burden of Neurodevelopmental Disorders? Journal of Childhood & Developmental Disorders. Vol. 4, 1-20. DOI: 10.4172/2472-1786.100073; http://childhood-developmental-disorders.imedpub.com/archive.php

 Answer

We thank the reviewer for the focus on the Cleaton paper. We have added the reference.

  1. Method

Unlikely, I do not have access to figure and tables so I did not control figure and tables.

Answer:

As explained it is very pitying the tables and figure were not accessible. They are now all included in the text.

Line 196, Reference 33 is already cited (number 26).

Line 198, reference is not correct.

Answer

The references were checked and corrected.

Line 210, something is missing in the sentence “Standard Error of Measurement …)

Answer:

Thank you for pointing this out. This seems to be an uploading issue, It is now corrected. See text

Text

The MABC Checklist investigates the effects of motor difficulties on activities of daily living. Standard Error of Measurement (SEM: 2 and 3 standard points) and Smallest Detectable Difference (four or more standard points) of MABC-2 total standard score.

In session 2.5 Intervention I would suggest to clarify better what saying at line 263, if I understood well the 18 children, who did NTT intervention, were trained in couple or maximum in four children. It is not crystallin clear to me.

Answer

I add the information about the group intervention size.

The intervention took place over nine weeks, with two sessions per week each lasting between 45 and 60 minutes. Each group consisted of 4 children. For more details on the content of the intervention, protocol sees appendix 1.

From line 268 to 272 a meagre description is provided for children who did Usual Care. Could the authors provide more details related to the kind of proposed activities in UC?

Answer:

We added more information about the UC

The UC group received regular physical education one time per week, including gymnastic exercises and sports-like activities. OT was also given once per week  and consisted of training of fine motor activities and sensorimotor therapy.

  1. Results

I restate that I do not have the possibility to see tables, however, text at line 305-308 has something missing. At line 308, the acronym “TSS” is not explained.

Probably, the sentence from line 318 to 320 should be insert after line 313.

Answer

The tables are now included and the missing text was again an upload problem. The acronym TSS

(Total Standard Score) is explained in the method section and in the tables.

  1. Discussion

In general discussion presents deep thoughts in relationship to the findings however, references, used to support the reasonings, need to be checked, for example, line 363 reference 44 is already reported as number 19.

Answer

References are checked and corrected.

From line 405 to 407, the authors speculate that the reason for the effect size medium to small after the NTT intervention could be due to the general difficulties in area of learning and its automation. However, another possibility could be the necessity of longer period of training to stabilize motor acquisitions.

Answer

I agree with the reviewer that this could be another explaining factor. We added a new sentence to enlighten this factor.

Another factor explaining the medium to small effect size can be the duration of the training period (9 weeks). It is expected that a longer duration will increase the possibility for a higher level of automation, however, may decrease the practicability and feasibility of the training in a school setting

From line 427, weaknesses of this study are illustrated. Since several times executive function problems are held to be cause of the complicated clinical profile of children with SLD/DCD, I would suggest to mention that no measurements related to executive function were reported in this study.

Answer:

We added this suggestion in the text

Another limitation is that no measures of specific executive functions of the children participating in this study were included. It is advised to add these in future research to have more information on the specific profile of the children regarding their executive function ability and the possible relation to the training.

At line 437, reference numbered 44 is reported before with number 19.

At line 462, reference of ICF is not indicated.

Reference needs to be checked from number 33 there are some mistakes:

-       reference 33 is already cited as 26;

-       reference 34 is not completed;

-       after reference 36, the paper of Smits-Engelman et al. (2013) is already cited as 23 so it needs to be cancelled;

-       reference 44 is cited as 19;

-       reference 48 and 50 need to start a new line

Answer

All references have been checked and adapted.

I hope the revision will clarify all the questions the reviewers addressed and improved the quality of the paper.

Best

Round 2

Reviewer 2 Report

Personally I think that the paper is now ready for publication because the authors have adequately addressed all of the issues raised.  I am not a native English speaker however it would be better to check the presence of some oversights though the manuscript.